# Predictive Model for Human Activity Recognition Based on Machine Learning and Feature Selection Techniques

**DOI:** 10.3390/ijerph191912272

**Published:** 2022-09-27

**Authors:** Janns Alvaro Patiño-Saucedo, Paola Patricia Ariza-Colpas, Shariq Butt-Aziz, Marlon Alberto Piñeres-Melo, José Luis López-Ruiz, Roberto Cesar Morales-Ortega, Emiro De-la-hoz-Franco

**Affiliations:** 1Department of Computer Science and Electronics, Universidad de la Costa CUC, Barranquilla 080002, Colombia; 2Department of Computer Science and IT, University of Lahore, Lahore 44000, Pakistan; 3Department of Systems Engineering, Universidad del Norte, Barranquilla 081001, Colombia; 4Department of Computer Science, University of Jaén, Campus Las Lagunillas, 23071 Jaén, Spain

**Keywords:** human activity recognition (HAR), machine learning, classification, feature selection

## Abstract

Research into assisted living environments –within the area of Ambient Assisted Living (ALL)—focuses on generating innovative technology, products, and services to provide medical treatment and rehabilitation to the elderly, with the purpose of increasing the time in which these people can live independently, whether they suffer from neurodegenerative diseases or disabilities. This key area is responsible for the development of activity recognition systems (ARS) which are a valuable tool to identify the types of activities carried out by the elderly, and to provide them with effective care that allows them to carry out daily activities normally. This article aims to review the literature to outline the evolution of the different data mining techniques applied to this health area, by showing the metrics used by researchers in this area of knowledge in recent experiments.

## 1. Introduction

The research area of assisted living environments (AAL) focuses on generating innovative technologies, products, and services to aid, medical care, and rehabilitation to elderly people with the purpose of increasing the time in which these people can live independently, whether they suffer from neurodegenerative diseases or disabilities. This key research area is responsible for the development of activity recognition systems (ARS), which are a valuable tool to identify the types of activities carried out by elderly people, and to provide them with effective assistance that allows them to carry out daily activities normally.

ARS are based on human activity recognition (HAR), which encompasses the recognition of a wide range of activities. Within these, this work focuses especially on activities of daily living (ADL). To evaluate the performance of ARS in the recognition of activities of daily living, it is necessary to use test data sets in experimental scenarios, which have been suitably designed by the scientific community for HAR.

Currently, a large part of the world’s elderly population suffers from neurodegenerative diseases. These types of diseases greatly affect the people who suffer from them, since they cause loss of balance, reduced mobility, speech deficiencies, breathing issues, and other alterations in cardiovascular function, which directly lead to a decrease in the cognitive abilities of individuals and, to a great extent, make it difficult to carry out activities of daily living [1]. Alzheimer’s, dementia, amyotrophic lateral sclerosis (ALS), and Parkinson’s are some of the most common types of neurodegenerative diseases.

However, before implementing these systems, it is necessary to evaluate their performance in the HAR process to optimize the classification of activities in indoor environments. In this project, a functional model for HAR was built, combining the logistic model trees (LMT) classification technique and the One R feature selection technique; from the latter, the 33 features that most improve the success rates of the model were identified. The metrics used to determine the quality level of the model were recall and precision, both at 95.90%.

This work is divided into five sections. This section introduces the work. Section 2 includes a review of the current state of the art in the field of HAR: applicability of these systems, types of recognition, types of data collection, and existing datasets in the literature. Section 3 details how the data was processed, how the functional model was built, and presents the different experimentation scenarios carried out in this research. In Section 4, the results obtained in the different scenarios are explained in detail. Finally, in Section 5, the conclusions obtained are presented and some future work is proposed.

## 2. Conceptual Information

### 2.1. Fundamentals Related to Human Activity Recognition

Currently, a large percentage of the world’s elderly population suffers from neurodegenerative diseases that affect not only memory, thought, and behavior, but also affect mobility, preventing the performance of certain daily activities [2]. These diseases have a significant effect on the quality of life, since those who are unable to carry out activities of daily living normally are forced to depend on others to try to lead a normal life and in some cases, they suffer from social isolation. The research area of Ambient Assisted Living (AAL) offers older adults various solutions to carry out these daily activities and live independently for as long as possible.

This key area is bringing about innovations based on information and communication technologies that provide aid, medical care, and rehabilitation to older people to improve their quality of life [3]. To do this, AAL solutions provide an ecosystem of sensors, computers, wireless networks, and software applications that allow monitoring of medical care with the main objective of making life easier and achieving a higher degree of independence for the elderly [4]. Among the products and services developed in the framework of AAL are activity recognition systems (ARS). Within such systems, one of the most important features to be implemented in AAL technologies are human activity recognition (HAR) processes.

### 2.2. Human Activity Recognition

HAR aims to identify the actions carried out by a person through a set of observations of the subject and the environment in which they operate [5]. Below we provide a review of the areas of application, types of data collection, and types of recognition. A summary of this content is provided in Table 1.

HAR is an area of research with numerous applications such as computer vision [6], video surveillance implemented in banks or airports, sports technique analysis, systems that allow interaction with video games through gestures and military tactics [7], in addition to assisted living environments providing care for the elderly or people with mental illness. This wide range of applications makes HAR a highly relevant and current research topic.

HAR recognizes patterns of human activity from different types of data, which are collected through different devices that contain a variety of sensors. For example (1) wearable devices that integrate accelerometers, gyroscopes, GPS, and heart rate sensors, among others; or (2) environmental sensors that collect numerical or categorical data directly from cameras that record image or video data. Thus, human activity recognition has been approached in two different ways in terms of the source or the type of device that collects the data: the first is wearable sensors which are directly attached to the user. The second source is external sensors, which are fixed by default to objects with which people will interact within a given area of interest [8].

Regarding human activities, in [6] these have been categorized or classified at different levels according to their complexity: gestures, actions, interactions, and group activities. Gestures are considered elementary movements of a part of the person’s body such as stretching an arm or lifting a leg. Actions are activities that can be composed of multiple gestures organized in a space of time performed by a person, such as walking or jumping. Interactions are human activities involving two or more people and/or objects; for example, two people fighting or one person doing the dishes. Finally, group activities involve multiple people and/or objects, such as a group hike or a fight between two groups.

Although the field of HAR is very broad, this research work focuses on the recognition of activities of daily living (ADL), which were defined in [9] as the set of activities that a person performs independently for their personal care, transport, and communication, such as personal mobility, eating, cleaning and resting, among others. Indeed, ARS based on HAR to recognize activities of daily living has the potential to bring significant improvements in the quality of life of people suffering from neurodegenerative diseases, but the performance of these systems must be previously tested and measured by evaluating various test data sets in experimental scenarios. To achieve this end, the scientific community has developed and promoted a variety of data collections available online, which contain information regarding activities of daily living, performed both in indoor and outdoor environments.

### 2.3. HAR Dataset

In the scientific literature in the field of ADL recognition, seven datasets [10] are highly referenced and the main characteristics of these are summarized below in Table 2.

The most relevant data sets are (1) the Van Kasteren dataset [11], which is a collection of binary values collected from a wireless sensor network (WSN) deployed in an enclosure occupied by two men; (2) the Kyoto [12], Aruba [13] and Multiresident [14] datasets, all of which are part of the CASAS project [12] carried out by WSU (Washington State University). The latter deployed a variety of environmental sensors in an apartment, which consisted of three bedrooms, a bathroom, a kitchen, and a living room.

For this study, we decided to evaluate the Aruba CASAS dataset, since it is a comprehensive dataset, whose raw files are available online on the official project site. Although it has been shown that the evaluation metrics [17] are 100% in terms of accuracy, in this study the second-best result so far was obtained, with an improvement in terms of accuracy and in computation times. This was achieved by evaluating other classification techniques and reducing the scope of the data by applying various feature selection techniques.

In this paper, the single and multiple occupancy dataset known as Aruba CASAS smart home project [13] from WSU (Washington State University) is used. This dataset collected different data sources in the home of an adult volunteer. The resident of the house was a woman who received visits from her children and grandchildren regularly between 4 November 2010 and 11 June 2011. Two data sources gave rise to the information, the first source was binary and was made up of movement and contact sensors, and the second source was made up of temperature sensors.

The binary source consisted of 35 sensors, of which 31 were movement sensors, identified by the letter M. These sensors were installed on the floor and detected the pressure exerted by the individual when stepping on the ground, representing the activation and deactivation states (ON/OFF). The remaining four (4) sensors were contact sensors, installed on the doors and identified by the letter D. These types of sensors detect the opening and closing states of the doors (OPEN/CLOSE). The second source was made up of 5 temperature sensors located in different places in the house and identified by the letter T. This type of sensor detects the temperature of the environment in continuous values represented in degrees Celsius.

The information contained in this dataset is made up of the recorded events, a product of the individual’s interactions with each of the sensors. For each event (each activity performed by the individual), the start and end date and time are recorded. In total, eleven activities were labeled, but in this study, only nine (9) were considered because the other two activities have a very low number of samples. For evaluation purposes, the following activities were considered: preparing meals (Meal_Preparation), resting (Relax), eating (Eating), working (Work), sleeping (Sleeping), going from bed to the bathroom (Bed_to_Toilet), getting home (Enter_Home), leaving home (Leave_Home) and cleaning (Housekeeping).

## 3. Building Predictive Models for HAR

This section describes the methodology applied: pre-processing of the datasets, aggregation functions, model building, and experimentation.

The proposal described here is based on the pre-processing of the original data [18,19,20] provided by the Aruba CASAS dataset. This resulted in the processed dataset, from which three new subsets of data were generated: Aruba CASAS–raw, Aruba CASAS–duration, and Aruba CASAS–sensor-based.

For each of the three datasets, the process of building a functional model was carried out, followed by a comparison of the quality metrics for each model and, finally, choosing the best-validated model and the correct configuration of the dataset in terms of feature categories.

This whole process is summarised in Figure 1.

### 3.1. Pre-Processing of Datasets

The starting point for this research was the original data provided by the Aruba CASAS dataset (detailed above), made up of the events recorded both by binary sensors (motion and contact) and by temperature sensors. In addition, it includes the start and end date and time of each activity. Initially, a pre-processing phase was carried out, which consisted in generating features from the representation of the activity duration time frames, extracted from the original data instances. This procedure gave rise to the processed dataset, whose structure is detailed below. The processed dataset is made up of a total of 69 features, divided into four (4) categories: count features, average features, aggregation features, and original features. The count features are built from the contact sensors in the doors. In total, there are four (4) contact sensors, and a count was made for both opening and closing (OPEN/CLOSE), within the duration frames of the activities. Therefore, eight (8) door contact sensor features were generated. Count features were also generated from the motion sensors. However, since motion sensors have nearly simultaneous ON and OFF states (i.e., an OFF state is executed immediately after the ON state), an event count was made from the pair of states (ON and OFF) for each sensor. Therefore, 31 motion sensor features were generated. Other count features are the number of events corresponding to a certain activity carried out in the time frame. The duration features represent the difference in seconds between the start date and time and the end date and time of the activity.

Regarding average features, these have been calculated from the temperature sensors, since the values they take are continuous data, adding a total of five (5) features to the dataset. Additionally, other aggregation features were generated from these sensors and, since four (4) statistical formulas were used (range, standard deviation, skew, and kurtosis) for each of the sensors, a total of 20 features were generated for this category. The three (3) remaining features are part of the category of original features and correspond to the class label, the start date and time, and the end date and time of the activity. For greater precision, Table 3 contains the structure of the processed dataset.

From the processed dataset, three data subsets were generated that will be called: Aruba CASAS–raw, Aruba CASAS–duration, and Aruba CASAS–sensor-based, which differ in the number of features and have the following configuration:−The Aruba CASAS–raw dataset has a total of 47 features, of which 39 correspond to the category of count features, five (5) to the category of average features, and the remaining three (3) to the category of original features.−The Aruba CASAS–duration dataset has a total of 49 features, of which 41 correspond to the category of count features, five (5) to the category of average features, and the remaining three (3) to the category of original features.−The Aruba CASAS–sensor-based dataset is made up of a total of 67 features, of which 39 correspond to the category of count features, five (5) to the category of average features, 20 to the category of aggregation features, and the remaining three (3) to the category of original features.

These datasets were generated to carry out subsequent tests and identify which dataset produces the best results in terms of the classification capacity of machine learning techniques. These techniques were proposed to evaluate the incidence of one or another category of features in the classification capacity of the technique. Table 4 identifies the number of features of these three datasets based on the categories of features that make them up.

Moreover, the data subsets used in the model construction process for training (train) and testing (test) follow the distribution of data instances presented in Table 5. The proportions correspond to 69.90% for the training subset and 30.10% for the testing subset, in each of the three datasets (Aruba CASAS–raw, Aruba CASAS–duration, and Aruba CASAS–sensor-based).

For the construction of each subset (training and testing), the instances were selected randomly, approximately the same proportion of instances for each class label. That is, approximately 70% for training and 30% for testing (see Table 6).

### 3.2. Aggregation Functions

To verify the Aruba CASAS–sensor-based dataset, it was necessary to calculate several features from aggregation functions. In this process, the instances were grouped by class criteria. That is, by activity, specifically from the temperature features, the functions used were: range, standard deviation, skewness, and kurtosis, using the functions defined in [21]. Each of these is detailed below:-*Range*: is the difference between the largest value and the smallest value in a data set.
(1)$$o_{range}=o_{max}-o_{min}\;$$

-*Standard deviation:* defined as the square root of the variance. The variance is the sum of all the squared differences of each occurrence value to the mean, divided by the number of sensors 𝑆 minus 1.


(2)
$$\sigma =\sqrt{\frac{1}{s-1}\overset{s}{\underset{i=1}{\sum }}\left(o_{i}-\mu^{2}\right)}\;$$


-*Skewness:* defined as the quotient of the third central moment 𝑚_3_ of a data set and the standard deviation cubed.


(3)
$$\gamma =\frac{m_{3}}{\sigma^{3}}=\frac{\frac{1}{s}\sum_{i=1}^{s}{\left(o_{i}-\mu \right)}^{3}}{{\sqrt{\frac{1}{s}\sum_{i=1}^{s}\left(o_{i}-\mu^{2}\right)}}^{3}}$$


-*Kurtosis:* defined as the quotient of the fourth central moment of a data set 𝑚_4_, and the standard deviation ***σ*** to the fourth power.


(4)
$$\kappa =\frac{m_{4}}{\sigma^{4}}=\frac{\frac{1}{s}\sum_{i=1}^{s}{\left(o_{i}-\mu \right)}^{4}}{{\sqrt{\frac{1}{s}\sum_{i=1}^{s}\left(o_{i}-\mu^{2}\right)}}^{4}}$$


### 3.3. Model Construction

Different models were built from the three datasets, implementing classification techniques, integrated with feature selection techniques. As a result of the evaluation of the quality metrics, the best results were identified for each dataset. That is, each evaluation made it possible to identify the best combinations of classification techniques with feature selection techniques, which generated the highest quality metrics for each evaluated dataset (Aruba CASAS–raw, Aruba CASAS–duration, and Aruba CASAS–sensor based).

A comprehensive comparative analysis of the results obtained by these evaluations made it possible to identify the dataset that generated the best classification results and the respective classification techniques and feature selection which led to those best results (see Figure 2).

### 3.4. Experimentation

We wanted to build a model that yields the best results in terms of quality metrics. In addition to evaluating different configurations of feature categories for the dataset, which have a major impact on the classification process, three experimentation scenarios were proposed. In our first experimental scenario, different classification techniques were applied to each of the three data subsets (Aruba CASAS–raw, Aruba CASAS–duration, and Aruba CASAS–sensor-based). Then, we wanted to identify the techniques that generate the best quality metrics in each of the experiments. For this evaluation, a random sampling of instances of each dataset was carried out to divide them into training and testing, of which each training data set (train) is 70% of the samples, and each testing data set (test) is approximately 30%.

In our second experimental scenario, different feature selection techniques were applied to the training and testing datasets of each data subset, and the optimal number of features was identified with the classification technique that best affects the evaluation process for each one of the data subsets. In our third experimental scenario, for each dataset, the performance of the best hybridization of classification technique with feature selection technique using 10-fold cross-validation was comprehensively evaluated. Each one of the proposed experimentation scenarios carried out for each data subset (Aruba CASAS–raw, Aruba CASAS–duration, and Aruba CASAS–sensor-based) is detailed below.

## 4. Experimentation Scenarios

Here we describe different experimentation scenarios for the creation of a predictive HAR (human activity recognition) model, applying different configurations of classification and feature selection techniques to the Aruba CASAS–raw, Aruba CASAS–duration, and Aruba CASAS–sensor-based data subsets generated from the original Aruba CASAS dataset. Subsequently, to compare the performance of different machine learning approaches, a comparative analysis of the quality metrics was performed on each of the three recreated scenarios: (1) with classification techniques, (2) through hybridization of classification and selection techniques, and (3) evaluating the best results through cross-validation. In the three scenarios, the three pre-processed data subsets were used to identify which data subset, when processed using the respective techniques, generates better quality metrics in the predictive process.

### 4.1. Experimental Scenario No. 1: Comparative Analysis of Classification Techniques on Data Subsets

In this first scenario, three experiments were carried out, each evaluating 31 classification techniques in the three datasets (Aruba CASAS–raw, Aruba CASAS–duration, and Aruba CASAS–sensor-based). For each experiment, subsets of data were used, from each dataset, for the training process (train) and the testing process (test). The classification techniques evaluated in the different experiments of this scenario are presented in Table 7, indicating the subcategory to which they correspond.

To define the features of the datasets, a series of feature selection algorithms were used in this experiment, which yielded the following results (see Table 8):

For the experiments with the Aruba CASAS–raw and Aruba CASAS–sensor-based datasets (see Table 8), the classifiers with the best results in terms of the recall metric were LMT with 94.50% and LogitBoost with 94.20% when both were evaluated. Additionally, it was possible to identify that in these cases the ROC area metric was 99.60% and 99.70%, respectively. Regarding the test with the Aruba CASAS–duration dataset, the classification techniques with the highest recall were J48 and JRIP at 95.60% for both classifiers, with JRIP presenting the highest ROC area metric at 99.30%. It is important to specify the implementation details of this classifier considering the feature selection process identified in Table 9. Table 10 shows the results of the LMT classifier using the GainRatio and OneR algorithms, respectively.

LMT is the classification technique that yields the best results in terms of recall with both the Aruba CASAS–raw and Aruba CASAS–sensor-based datasets. Regarding the Aruba CASAS–duration dataset, even though LMT was not the technique with the best classification results, it reached a Recall of 95.40%, as shown in Table 11 below.

### 4.2. Experimental Scenario No. 2: Comparative Analysis of the Hybridization of Selection and Classification Techniques on Data Subsets

In this scenario, three experiments were carried out, each with the respective datasets mentioned above (Aruba CASAS–raw, Aruba CASAS–duration, and Aruba CASAS–sensor-based). To minimize computation times, we sought to reduce the size of the three datasets, identifying the set of features that best affect the classification. For this purpose, the Info Gain [44], Gain Ratio [44], Symmetrical Uncert [44], OneR [30], and Relief feature selection techniques [45] were combined with each of the four classification techniques. The results were then analyzed for each scenario and each dataset to see which generated better results.

Once the quality metrics were evaluated, it was possible to determine that the hybridization of the classification techniques with the feature selection techniques which generated the best results were: (1) LMT with Gain Ratio using both 27 and 24 features for the Aruba CASAS dataset–raw (see Table 12); (2) JRIP with One R using 47 features and LMT with One R using 33 features for the Aruba CASAS–duration dataset (see Table 13); and (3) LMT with Info Gain using 47 features and LMT with Gain Ratio using 31 features for the Aruba CASAS–sensor-based dataset (see Table 14).

In this scenario, the application of different feature selection techniques was carried out to select the features to be selected to be included in the experimentation (see Table 15):

For the experiment with the Aruba CASAS–raw dataset, both proposals (27 and 24 features) managed to increase recall to 94.90% and ROC area to 99.70% (compared with the initial scenario in which the datasets with all 47 features had a recall of 94.50% and a ROC area of 99.60%). LMT with Gain Ratio (24 features) achieved a greater reduction in the number of attributes used in the classification process (see Table 9).

In the experiment with the Aruba CASAS–duration dataset, even though the J48 classifier (in the first experimentation scenario) had generated very good results, reaching 95.60% recall (with 49 features, as can be seen in Table 3), the hybridization proposals JRIP with One R using 47 features and LMT with One R using 33 features (and executed in this second scenario), increased recall, reaching 95.80% and 95.90% respectively. In addition, a significant reduction in the number of features was achieved for the classification process. For a better overview of this (see Table 13). It is evident that of these two combinations of techniques, it is better to use LMT with One R because it generates greater recall (95.90%) and because it only requires 33 features for the classification process.

In the first scenario for the Aruba CASAS–sensor-based dataset with 67 features, a recall of 94.50% and a ROC area of 99.70% was obtained using the LMT technique. In this scenario, with the same dataset, both proposals (LMT + Info Gain with 47 features and LMT + Gain ratio with 31 features) showed an increase in recall, which was 94.90%. LMT with Gain Ratio was the combination that achieved a greater decrease in the number of features (which affects the computation time required by the predictive model), as can be seen in Table 14.

Let us compare the two best hybridizations for each dataset:-In the Aruba CASAS–raw dataset, the two combinations presented the same results in terms of recall, F-Measure, and ROC area. LMT with Gain Ratio using 24 features presented the lowest FP-Rate at 0.5%.-In the evaluation of the Aruba CASAS–duration dataset, the combination with the best recall (95.90%) and ROC area (99.70%) was LMT with One R, using 33 features.-Regarding the evaluation of the Aruba CASAS–sensor-based dataset, the results for the two hybridizations of classification and selection techniques used coincided with the respective results of the precision, recall, F-Measure, and ROC area metrics.

Although LMT with Gain Ratio for 31 features is the combination that presented the highest FP Rate of 0.6%, it is important to highlight that the other combination (LMT with Info Gain) uses 16 more features (see Table 16). Until this point it can be deduced that the dataset that generates the best predictive model is Aruba CASAS–duration, after applying the hybridization of LMT techniques with One R, using only 33 features of the 49 original features. In this order of ideas, these 33 features have the best effect on the classification process to predict human activities. Table 17 indicates the priority of incidence in the prediction identified from the One R selection technique.

### 4.3. Experimental Scenario No. 3: Comparative Analysis of the Best Results Obtained, Applying Cross-Validation

In this scenario, a more exhaustive evaluation was carried out to assess whether there is a better combination of classification techniques and feature selection, compared to the previous scenario, for each dataset (Aruba CASAS–raw, Aruba CASAS–duration, and Aruba CASAS-sensor-based). Each dataset was trained and tested using 10-fold cross-validation, generating three experiments, the results of which are detailed in Table 18, Table 19 and Table 20.

In the cross-validation process, each complete dataset was divided into 10 folds of equal size. Iterative tests were then performed in which the model was trained on 9 folds and tested on the remaining fold. Finally, the quality metrics obtained in each of the 10 iterations were averaged to calculate the result. For the test with the Aruba CASAS–raw dataset, with the LMT classification technique and Gain Ratio feature selection (24 features), the recall was 94.10% (see Table 18). This is not an improvement over the results of the second scenario (same dataset and same combination of techniques), where recall was 94.90% (see Table 12).

In the test with the Aruba CASAS–duration dataset, with the LMT classification technique and One R feature selection (33 features), the recall was 94.10% (see Table 15). This is not an improvement over the evaluation carried out for this dataset with the same combination of techniques in the second scenario, where recall was 95.90% (see Table 15).

Regarding the test with the Aruba CASAS–sensor-based dataset, with the LMT classification technique and One R feature selection (31 features), the recall was 94.00% (see Table 18). This is also not an improvement over the evaluation carried out for this dataset with the same combination of techniques in the second scenario, where recall was 94.90% (see Table 18).

The results obtained in this third experimentation scenario, in terms of recall and ROC area after applying cross-validation, did not show improvements compared to those obtained in the second scenario. This behavior occurred in each of the experiments carried out with the datasets (Aruba CASAS–raw, Aruba CASAS–duration, and Aruba CASAS–sensor-based) due to overfitting (see Table 21). Overfitting is the result of over-training a model with data adjusted to specific features of the dataset. That is, excessive learning of certain class behaviors means that, in turn, the understanding of behaviors that are different from the class label is impossible. According to [46], this is a result of an imbalance in the training data set.

To check if there are significant differences between the proposed models, a statistical analysis was carried out through the study of their variance. For this, the null hypothesis Ho was proposed, which posits equality between the means of the models with an alpha level of significance of 5%, and an alternative hypothesis H1 rejects said equality.

In Table 22, the probability values for the three models—M1 (LMT + Gain Ratio 24 features) vs. M2 (LMT + One R 33 features), M1 (LMT + Gain Ratio 24 features) vs. M3 (LMT + Info Gain 47 features) and M2 (LMT + One R 33 features) vs. M3(LMT + Info Gain 47 features)—are much higher than the 5% alpha level of significance. So, the null hypothesis Ho was accepted, which poses the equality between the means of the models. This indicates that there is no significant difference between the three models proposed, in addition to the consistency of the data considered for the experimentation.

## 5. Conclusions

In this section, the results and conclusions reached with the development of this research work are presented, after evaluating each experimentation scenario previously proposed.

In the first scenario, the recall quality metric of 95.60% represents the best result, when the Aruba CASAS–duration dataset was evaluated using 49 features and the J48 and JRIP classification techniques. This surpassed the results of the Aruba CASAS–raw and Aruba CASAS–sensor-based datasets, in which a recall of 94.50% was obtained for both. This shows that adding the two count features for the number of events and activity duration improved recall by 1.1%. On the other hand, the Aruba CASAS–sensor-based dataset did not show any improvements over the Aruba CASAS–duration dataset. On the contrary, the results show an increase in computation times during the classification process. Aruba CASAS–sensor-based dataset has 20 additional features calculated through the aggregation functions applied to the features, generated from the temperature sensors, and which have been calculated by grouping the instances of the original dataset, segmented by classes—activities.

In the second scenario, again the experiment with the best result in terms of recall was the Aruba CASAS–duration dataset. The hybridization of the LMT classification technique with the One R feature selection technique, using 33 features, reached a recall of 95.90% compared to 95.80% achieved by JRIP and One R using 47 features. Additionally, the hybridization of LMT and One R achieved a significant reduction (of 32.65%) in the number of features (16 fewer features) compared to just 4.08% (a reduction of two features) achieved by the hybridization of JRIP and One R. Thus, using the combination of LMT and One R is used will have a direct impact in terms of decreasing in the computation times required for the construction and evaluation of the predictive model.

On the other hand, in the second scenario, regarding the experiments with the Aruba CASAS–raw and Aruba CASAS–sensor-based datasets, there was also a significant reduction in the number of features. Specifically, with the hybridization of LMT and Gain Ratio, using 24 features, for Aruba CASAS–raw and the hybridization of LMT and Gain Ratio, using 31 features, for Aruba CASAS–sensor-based. Although the decrease in the number of features is 48.94% and 53.73%, respectively, recall is 1.00% lower than the value obtained in the Aruba CASAS–duration dataset experiment. It is important to highlight that the classification technique that yielded the best results for each of the experiments with the three datasets, in terms of quality metrics, was the LMT technique. Table 12 presents the ranking of the 33 features that most affect the classification process, as determined by the One R feature selection technique.

In the third scenario, a comprehensive evaluation was carried out to determine the best hybridization for each dataset using 10-fold cross-validation. Here, a decrease in recall was found in each of the experiments with the three datasets (Aruba CASAS–raw, Aruba CASAS–duration, and Aruba CASAS–sensor-based) due to overfitting.

Regarding the recall quality metric corresponding to each class label (activity) in each of the experiments, it should be noted that the winning hybridization for the Aruba CASAS–duration dataset, despite having yielded a low 83.50% for the “Leave_Home” activity, managed to surpass the 55.60% achieved in both cases by the winning hybridizations of the Aruba CASAS–raw and Aruba CASAS–sensor-based dataset by 28.24%. This may be due to the inclusion of the 2 additional features for the number of events and duration of the activity, including for the Aruba CASAS–duration dataset, given that particularly for said activity the number of events (readings of sensors) is very low (see Table 21).

The recall metric for the cleaning activity (Housekeeping) has yielded different values in the experiments with each dataset. Despite having achieved 100.00% with the Aruba CASAS–raw dataset, its result with the other two datasets was not the best: in Aruba CASAS–duration it was 77.80% and in Aruba CASAS–sensor-based it was 88.90%. The difference in the results obtained for recall in each dataset was due to the low number of instances of this activity compared to the others, just 32 data instances (see Table 6). The highest success rates in terms of quality metrics were obtained when training the model with the Aruba CASAS–duration dataset, obtaining 95.90% in the recall, which indicates a high proportion of positive cases. This is a high detection rate for activities that were correctly identified. The 99.70% reached in the ROC area indicates that the model has very high predictive quality (see Table 18). In addition, there was a very low average detection rate of false positives with an FP rate of 0.60%. The average accuracy of 95.90% was also reached, which indicates that there is a high proportion of correct predictions, both positive and negative, in the total number of predictions. An F-Measure of 95.80% was also reached (see Table 23).

Consequently, the model proposed in this research integrates the LMT classification technique with the One R feature selection technique, using only 33 of the 49 features available in the Aruba CASAS—duration dataset for human activity recognition: preparing meals (Meal_Preparation), resting (Relax), eating (Eating), working (Work), sleeping (Sleeping), going from bed to the bathroom (Bed_to_Toilet), getting home (Enter_Home), leaving home (Leave_Home) and cleaning (Housekeeping). Said data was collected from an indoor environment by the WSU (Washington State University) smart home project.

Finally, this research work makes two important contributions to the area of human activity recognition (HAR): firstly, the pre-processing of the original Aruba CASAS dataset provided by the WSU smart home project, which is available in an online repository with all its raw records. Finally, the identification of the classification and feature selection techniques that yield the best metrics by class criterion, is based on the construction of a model that evaluates said dataset.

## Figures and Tables

**Figure 1 ijerph-19-12272-f001:**
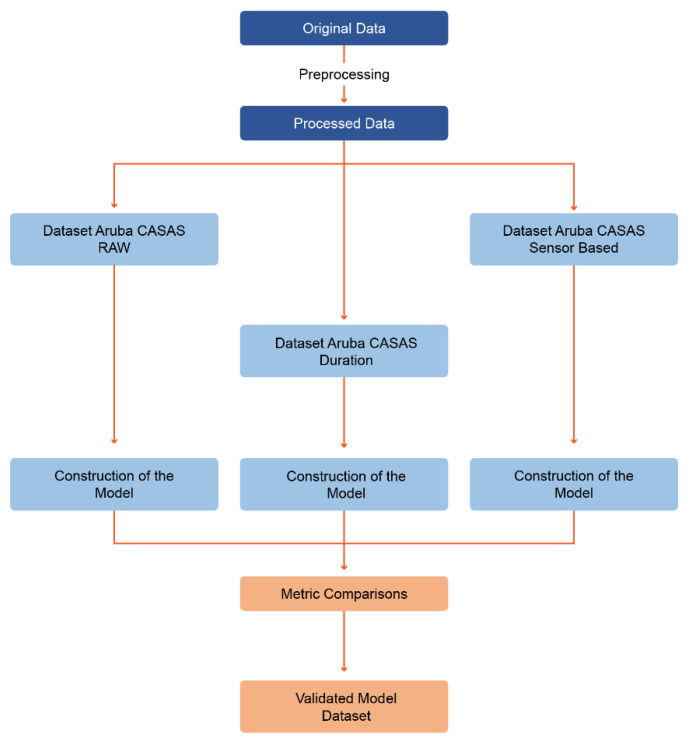
Data preparation and construction of the proposed model.

**Figure 2 ijerph-19-12272-f002:**
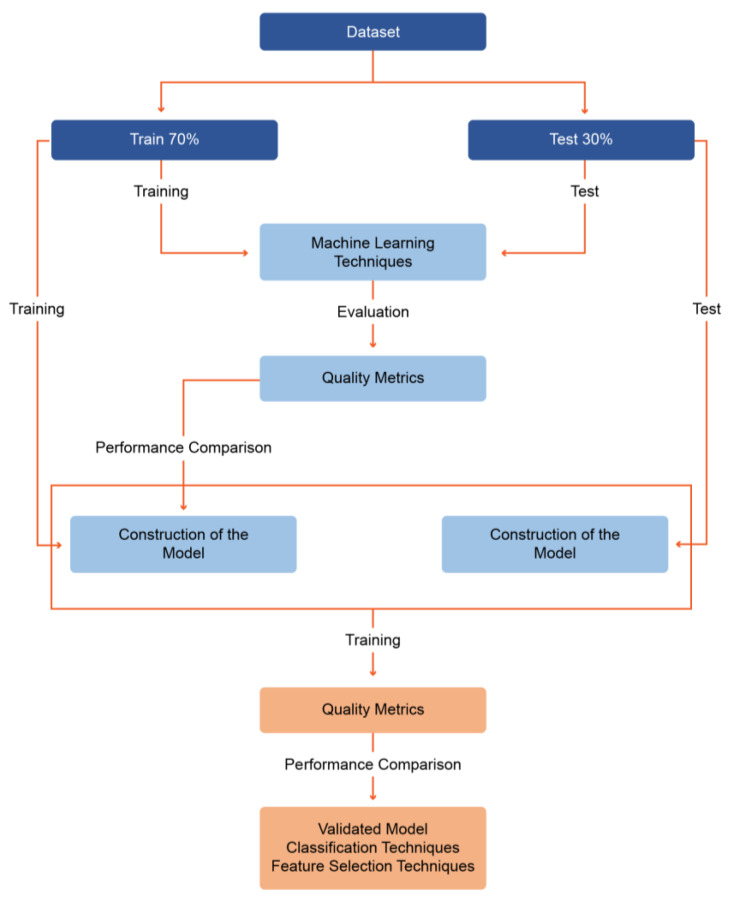
Model construction.

**Table 1 ijerph-19-12272-t001:** Summary of HAR features.

Data Collection Type [6]		Recognition Type [7,8]	Application Areas [7,9]
Wearable devices and sensors: accelerometers, gyroscopes, GPS, electrocardiogram, magnetometer, and heart rate, among others	Environmental sensors: binary sensors and cameras	Gestures, actions, interactions, and activities (e.g., daily living)	Computer vision, video surveillance (e.g., banks or airports), sport technique analysis, interaction with video games through gestures, military tactics, assisted living environments for health care of the elderly people or other diseases

**Table 2 ijerph-19-12272-t002:** Main datasets for ADL and their main characteristics.

Dataset	Event	Occupancy	Devices	Datatype	Context
Van Kasteren [11]	Activities	Single	Wireless Sensor Network and sensors	Binary values	Two houses (kitchen and bathroom)
CASAS Kyoto [12]	Activities	Single	Motion, associated with objects and telephone sensors	Datetime, sensor id, and value (binary or numerical)	Washington State University smart workplace
CASAS Aruba [13]	Activities	Multi-occupancy	Motion, door, and temperature sensors	Datetime, sensor id, and value (binary or numerical)	Washington State University smart workplace
CASAS Multiresident [14]	Activities	Multi-occupancy	Motion, item, cabinet, water, burner, phone, and temperature sensors	Datetime, sensor id, value (binary), inhabitant id, and task id	Washington State University smart apartment
UCI HAR [5]	Actions	N/A	Accelerometer and gyroscope	Normalized values between [−1, 1]	Handset mounted: on the left side of the belt and placed according to the user’s preference
Opportunity [15]	Activities	Interleaved and hierarchical naturalistic activities	Inertial sensors and accelerometers	Text file (array, each row is a sample)	A room simulating a studio flat
mHealth [16]	Actions	N/A	Accelerometer, electrocardiogram, gyroscope, and magnetometer	Fine-grained real-valued sensor readings of actions over a short time interval	Chest, wrist, and ankle

**Table 3 ijerph-19-12272-t003:** Processed dataset structure.

	Count Features				Average Features	Aggregation Features	Original Features			Total
	Door Contact Sensors	Motion Sensors	Number of Events	Duration of Activity	Thermometers	Thermometers	Start of Activity	End of Activity	Class	
**Features**	D001-open, D001-close, D002-open, D002-close, D003-open, D003-close, D004-open y D004-close (Total: 8)	M001, M002, M003, M004, M005, M006, M007, M008, M009, M010, M011, M012, M013, M014, M015, M016, M017, M018, M019, M020, M021, M022, M023, M024, M025, M026, M027, M028, M029, M030 y M031 (Total: 31)	Events (Total: 1)	Duration (Total: 1)	T001, T002, T003, T004 Y T005 (Total: 5)	T001-RANGE, T001-DESV, T001-BIAS, T001-KURT, T002-RANGE, T002-DESV, T002-BIAS, T002-KURT, T003-RANGE, T003-DESV, T003-BIAS, T003-KURT, T004-RANGO, T004-DESV, T004-BIAS, T004-KURT, T005-RANGE, T005-DESV, T005-BIAS y T005-KURT (Total: 20)	Start date and time (Total: 1)	Start date and time (Total: 1)	Activity (Total: 1)	**69 Features**

**Table 4 ijerph-19-12272-t004:** Dataset configuration Aruba CASAS–raw, Aruba CASAS–duration, and Aruba CASAS–sensor-based.

Dataset	Count Features				Average Features	Aggregation Features	Original Features			Number of Features
	Door Contact Sensors	Motion Sensors	Number of Events	Duration of Activity	Thermometers	Thermometers	Start of Activity	End of Activity	Class	
**Aruba CASAS** **–** **raw**	8	31	0	0	5	0	1	1	1	**47**
**Aruba CASAS** **–** **duration**	8	31	1	1	5	0	1	1	1	**49**
**Aruba CASAS** **–** **sensor-based**	8	31	0	0	5	20	1	1	1	**67**

**Table 5 ijerph-19-12272-t005:** Distribution of instances for training and testing data subsets of the Aruba CASAS–raw, Aruba CASAS–duration, and Aruba CASAS–sensor-based datasets.

Dataset	Training Subset		Testing Subset		Total Instances
	Instances	Percentage	Instances	Percentage	
**Aruba CASAS** **–** **raw**	4460	69.9%	1916	30.1%	6376
**Aruba CASAS** **–** **duration**	4460	69.9%	1916	30.1%	6376
**Aruba CASAS** **–** **sensor-based**	4460	69.9%	1916	30.1%	6376

**Table 6 ijerph-19-12272-t006:** Distribution of data instances by class for training and testing data subsets for the Aruba CASAS–raw, Aruba CASAS–duration, and Aruba CASAS–sensor-based datasets.

Dataset/Class		Sleeping	Bed to Toilet	Meal Preparation	Relax	House Keeping	Eating	Leave Home	Enter Home	Work
**Aruba CASAS** **–** **raw**	**Train**	265	112	1105	2036	23	181	298	319	121
	**Test**	136	45	482	878	9	71	133	112	50
**Aruba CASAS** **–** **duration**	**Train**	265	112	1105	2036	23	181	298	319	121
	**Test**	136	45	482	878	9	71	133	112	50
**Aruba CASAS** **–** **sensor-based**	**Train**	265	112	1105	2036	23	181	298	319	121
	**Test**	136	45	482	878	9	71	133	112	50

**Table 7 ijerph-19-12272-t007:** Techniques evaluated in the different experiments by subcategories.

Subcategories	Technique	Function
**Decision Tree**	Logistic Model Trees—LMT [22]	Build logistic model trees.
	J48 (C4.5 decision tree) [23]	Decision tree based on algorithm C4.5.
	Reduced-Error Pruning Tree—REPTree [24]	Fast tree learning using pruning in error reduction.
	RandomForest [25]	Construction of random trees
	Random Tree [24]	Build a tree that considers a random number of given features at each node.
	DecisionStump [26]	Build one-level decision trees
**Rules**	JRip [27]	RIPPER (Reduced Incremental Pruning to Produce Error Reduction) algorithm for fast, efficient rule induction.
	Partial Decision Trees—PART [24]	Obtains rules from decision trees built using J4.8.
	Decision Table [28]	Construct a simple decision table for the majority classifier.
	ZeroR, Stacking [29]	Predict the majority class (if nominal) or the average value (if numeric).
	OneR [30]	One rule classifier
**Functions**	Logistic [31]	Build linear logistic regression models.
	MultilayerPerceptron [32]	Backpropagation Neural Network
**Multiclassifiers (Meta)**	Random Committee [24]	Build a set of random base classifiers
	Stacking [29]	Combine multiple classifiers using the stacking method.
	LogitBoost [33]	Perform additive logistic regression
	Classification Via Regression [24]	It performs classification using a regression method
	MultiClass Classifier [34]	Use a two-class classifier for multiclass data sets
	Bagging [35]	A bag classifier works by regression as well.
	AdaBoostM1 [36]	Use the AdaBoostM1 method.
	Vote [37]	Combine classifiers using average probability estimates or numerical predictions
	CVParameterSelection [38]	Performs parameter selection through cross-validation
	MultiScheme [39]	Uses cross-validation to select a classifier from multiple candidates
	AttributeSelectedClassifier [24]	Reduces the dimensionality of the data by selecting attributes.
	RandomSubSpace [40]	Build a decision tree-based classifier that maintains the highest accuracy on the training data.
	Filtered Classifier [39]	Run a classifier on filtered data
**Lazy algorithms**	IB1 Instance-based Learning Algorithms [41]	Instance-based learning is a basic nearest neighbor
	IB2 Instance-based Learning Algorithms [41]	K nearest neighbor classifier.
	IB3 Instance-based Learning Algorithms [41]	K nearest neighbor classifier.
	KStar [42]	A nearest neighbor with a generalized distance function
	LWL [43]	A general algorithm for locally heavy learning.

**Table 8 ijerph-19-12272-t008:** Feature Prioritization Algorithms approach.

Algorithm	Feature Prioritization
GainRatio	12, 11, 9, 8, 35, 25, 18, 28, 13, 27, 26, 24, 36, 39, 16, 40, 14, 22, 23, 37, 29, 34, 30, 38, 19, 10, 15, 44, 43, 41, 42, 31, 2, 45, 17, 21, 5, 4, 20, 33, 32, 1, 0, 7, 3, 6, 46, 47
InfoGain	18, 27, 28, 24, 22, 26, 29, 9, 25, 8, 23, 39, 43, 16, 42, 12, 44, 45, 11, 41, 30, 19, 14, 13, 35, 38, 36, 37, 31, 21, 15, 17, 40, 34, 33, 1, 0, 32, 10, 5, 20, 4, 2, 6, 7, 3, 46, 47
OneR	27, 28, 24, 18, 26, 25, 22, 23, 30, 9, 8, 39, 16, 12, 11, 29, 43, 38, 31, 35, 13, 42, 14, 36, 37, 41, 17, 44, 45, 15, 32, 40, 5, 4, 34, 10, 2, 6, 19, 7, 21, 3, 20, 33, 0, 1, 46, 47
ReliefF	44, 43, 42, 18, 9, 28, 27, 8, 41, 24, 26, 12, 25, 39, 22, 23, 29, 38, 13, 35, 40, 31, 36, 16, 37, 19, 30, 11, 45, 14, 4, 5, 15, 33, 17, 32, 21, 34, 10, 3, 6, 1, 7, 0, 2, 20, 46, 47

**Table 9 ijerph-19-12272-t009:** LMT with GainRatio approach.

TP Rate	FP Rate	Precision	Recall	F-Measure	MCC	ROC Area	PRC Area	Class
1.000	0.000	1.000	1.000	1.000	1.000	1.000	1.000	Sleeping
1.000	0.000	1.000	1.000	1.000	1.000	1.000	1.000	Bed_to_Toilet
0.983	0.012	0.963	0.983	0.973	0.964	0.996	0.981	Meal_Preparation
0.998	0.005	0.994	0.998	0.996	0.993	1.000	1.000	Relax
0.889	0.000	1.000	0.889	0.941	0.943	1.000	0.989	Housekeeping
0.986	0.003	0.933	0.986	0.959	0.958	1.000	0.987	Eating
0.000	0.001	0.000	0.000	0.000	−0.002	0.927	0.103	Wash_Dishes
0.541	0.015	0.727	0.541	0.621	0.604	0.981	0.698	Leave_Home
0.759	0.033	0.582	0.759	0.659	0.641	0.979	0.644	Enter_Home
1.000	0.000	1.000	1.000	1.000	1.000	1.000	1.000	Work
**0** **.939**	**0** **.008**	**0** **.934**	**0** **.939**	**0** **.935**	**0** **.929**	**0** **.996**	**0** **.945**	**Weighted Avg**

**Table 10 ijerph-19-12272-t010:** LMT with OneR approach.

TP Rate	FP Rate	Precision	Recall	F-Measure	MCC	ROC Area	PRC Area	Class
1.000	0.000	1.000	1.000	1.000	1.000	1.000	1.000	Sleeping
1.000	0.000	1.000	1.000	1.000	1.000	1.000	1.000	Bed_to_Toilet
0.983	0.012	0.965	0.983	0.974	0.966	0.995	0.978	Meal_Preparation
0.999	0.005	0.994	0.999	0.997	0.994	1.000	0.999	Relax
0.889	0.001	0.889	0.889	0.889	0.888	1.000	0.967	Housekeeping
0.986	0.002	0.959	0.986	0.972	0.971	1.000	0.987	Eating
0.000	0.001	0.000	0.000	0.000	-0.003	0.923	0.098	Wash_Dishes
0.564	0.018	0.701	0.564	0.625	0.605	0.980	0.692	Leave_Home
0.714	0.032	0.580	0.714	0.640	0.619	0.978	0.639	Enter_Home
1.000	0.000	1.000	1.000	1.000	1.000	1.000	1.000	Work
**0** **.939**	**0** **.008**	**0** **.933**	**0** **.939**	**0** **.935**	**0** **.929**	**0** **.995**	**0** **.944**	**Weighted Avg**

**Table 11 ijerph-19-12272-t011:** Techniques evaluated in the different experiments by subcategories.

Dataset	Quality Metrics					Classification Technique
	FP Rate	Precision	Recall	F-Measure	ROC Area	
**Aruba CASAS** **—** **raw**	0.50%	94.80%	94.50%	**94.50%**	**99.60%**	**LMT**
	0.60%	94.60%	94.20%	94.00%	99.70%	LogitBoost
	0.60%	94.30%	94.10%	94.10%	99.70%	ClassificationViaRegression
	0.50%	94.30%	94.00%	94.00%	99.00%	J48
**Aruba CASAS** **—** **duration**	0.50%	95.70%	95.60%	95.60%	99.00%	J48
	0.60%	95.70%	95.60%	95.50%	99.30%	JRIP
	0.60%	95.40%	95.40%	**95.40%**	**99.60%**	**LMT**
	0.70%	95.20%	95.30%	95.10%	99.80%	RandomSubSpace
**Aruba CASAS** **—** **sensor based**	0.70%	94.80%	94.50%	**94.40%**	**99.70%**	**LMT**
	0.60%	94.60%	94.20%	94.00%	99.70%	LogitBoost
	0.60%	94.30%	94.10%	94.10%	99.70%	ClassificationViaRegression
	0.60%	94.20%	93.90%	93.90%	99.00%	J48

**Table 12 ijerph-19-12272-t012:** Comparison between the hybridization of LMT + Gain Ratio techniques with a different number of features for the Aruba CASAS–raw training and testing datasets.

Class	LMT + Gain Ratio (27 Features)					LMT + Gain Ratio (24 Features)				
	FP Rate	Precision	Recall	F-Measure	ROC Area	FP Rate	Precision	Recall	F-Measure	ROC Area
**Sleeping**	0.00%	100.00%	100.00%	100.00%	100.00%	0.00%	100.00%	100.00%	100.00%	100.00%
**Bed_to_Toilet**	0.00%	100.00%	100.00%	100.00%	100.00%	0.00%	100.00%	100.00%	100.00%	100.00%
**Meal_Preparation**	0.10%	99.80%	98.50%	99.20%	99.90%	0.10%	99.60%	98.50%	99.10%	99.80%
**Relax**	0.60%	99.30%	99.80%	99.50%	100.00%	0.40%	99.50%	99.40%	99.50%	99.90%
**Housekeeping**	0.00%	100.00%	88.90%	94.10%	100.00%	0.10%	90.00%	100.00%	94.70%	100.00%
**Eating**	0.20%	94.70%	100.00%	97.30%	100.00%	0.30%	92.20%	100.00%	95.90%	100.00%
**Leave_Home**	1.50%	73.00%	54.90%	62.70%	98.10%	1.50%	73.30%	55.60%	63.20%	98.10%
**Enter_Home**	3.30%	59.00%	75.90%	66.40%	97.90%	3.20%	59.40%	75.90%	66.70%	97.90%
**Work**	0.00%	100.00%	100.00%	100.00%	100.00%	0.00%	100.00%	100.00%	100.00%	100.00%
**Average**	0.60%	95.20%	**94** **.** **90%**	94.90%	**99** **.** **70%**	0.50%	95.10%	**94** **.** **90%**	94.90%	**99** **.** **70%**

**Table 13 ijerph-19-12272-t013:** Comparison between the hybridization of JRIP + One R and LMT + One R techniques with the Aruba CASAS–duration training and testing datasets.

Class	JRIP + One R (47 Features)					LMT + One R (33 Features)				
	FP Rate	Precision	Recall	F-Measure	ROC Area	FP Rate	Precision	Recall	F-Measure	ROC Area
**Sleeping**	0.00%	100.00%	98.50%	99.30%	99.60%	0.00%	100.00%	100.00%	100.00%	100.00%
**Bed_to_Toilet**	0.00%	100.00%	97.80%	98.90%	98.90%	0.00%	100.00%	97.80%	98.90%	100.00%
**Meal_Preparation**	0.30%	99.20%	98.60%	98.90%	99.20%	0.10%	99.80%	98.60%	99.20%	99.90%
**Relax**	0.50%	99.40%	99.50%	99.50%	99.60%	0.70%	99.20%	99.70%	99.40%	99.80%
**Housekeeping**	0.10%	87.50%	77.80%	82.40%	88.90%	0.00%	100.00%	77.80%	87.50%	100.00%
**Eating**	0.40%	90.90%	98.60%	94.60%	98.80%	0.30%	92.10%	98.60%	95.20%	98.80%
**Leave_Home**	2.20%	73.50%	81.20%	77.10%	98.30%	2.30%	72.50%	83.50%	77.60%	98.50%
**Enter_Home**	1.30%	75.30%	65.20%	69.90%	97.00%	1.30%	75.30%	62.50%	68.30%	98.20%
**Work**	0.10%	98.00%	100.00%	99.00%	100.00%	0.10%	98.00%	96.00%	97.00%	100.00%
**Average**	0.50%	95.80%	**95** **.** **80%**	95.80%	**99** **.20%**	0.60%	95.90%	**95** **.** **90%**	95.80%	**99** **.70%**

**Table 14 ijerph-19-12272-t014:** Comparison between the hybridization of LMT + Info Gain and LMT + Gain Ratio techniques with the Aruba CASAS–sensor-based testing and training datasets.

Class	LMT + Info Gain (47 Features)					LMT + Gain Ratio (31 Features)				
	FP Rate	Precision	Recall	F-Measure	ROC Area	FP Rate	Precision	Recall	F-Measure	ROC Area
**Sleeping**	0.10%	99.30%	100.00%	99.60%	100.00%	0.00%	100.00%	100.00%	100.00%	100.00%
**Bed_to_Toilet**	0.00%	100.00%	100.00%	100.00%	100.00%	0.00%	100.00%	100.00%	100.00%	100.00%
**Meal_Preparation**	0.10%	99.80%	99.20%	99.50%	100.00%	0.20%	99.40%	98.30%	98.90%	100.00%
**Relax**	0.50%	99.40%	99.80%	99.60%	99.90%	0.60%	99.30%	99.80%	99.50%	100.00%
**Housekeeping**	0.00%	100.00%	44.40%	61.50%	91.00%	0.00%	100.00%	88.90%	94.10%	100.00%
**Eating**	0.20%	94.70%	100.00%	97.30%	100.00%	0.20%	94.60%	98.60%	96.60%	100.00%
**Leave_Home**	1.50%	73.00%	54.90%	62.70%	98.00%	1.50%	73.30%	55.60%	63.20%	98.10%
**Enter_Home**	3.30%	59.00%	75.90%	66.40%	97.80%	3.20%	59.40%	75.90%	66.70%	97.90%
**Work**	0.10%	98.00%	100.00%	99.00%	100.00%	0.00%	100.00%	100.00%	100.00%	100.00%
**Average**	0.50%	95.10%	**94** **.** **90%**	94.80%	**99** **.70%**	0.60%	95.10%	**94** **.** **90%**	94.80%	**99** **.70%**

**Table 15 ijerph-19-12272-t015:** Feature Prioritization Algorithms approach.

Algorithm	Feature Prioritization
GainRatio	13, 14, 11, 10, 37, 27, 20, 30, 15, 29, 26, 28, 38, 41, 18, 42, 16, 24, 25, 39, 2, 31, 3, 36, 32, 40, 21, 12, 17, 46, 43, 44, 33, 45, 4, 47, 19, 23, 7, 6, 22, 35, 34, 1, 0, 8, 5, 9, 48, 49
InfoGain	2, 20, 29, 30, 26, 3, 24, 28, 31, 11, 27, 10, 25, 41, 45, 18, 44, 14, 46, 47, 13, 43, 32, 21, 16, 15, 37, 40, 38, 39, 33, 23, 17, 19, 42, 36, 35, 1, 0, 34, 12, 7, 22, 6, 4, 9, 5, 8, 48, 49
OneR	29, 30, 26, 20, 28, 2, 27, 24, 25, 32, 3, 11, 10, 41, 18, 14, 31, 13, 45, 40, 46, 44, 37, 15, 43, 33, 16, 38, 47, 39, 19, 17, 34, 42, 7, 36, 6, 12, 4, 23, 8, 21, 9, 35, 5, 22, 1, 0, 48, 49
ReliefF	46, 45, 44, 20, 11, 30, 3, 29, 10, 43, 26, 28, 27, 14, 41, 24, 2, 25, 31, 40, 15, 37, 42, 33, 38, 18, 39, 21, 32, 13, 47, 16, 6, 7, 35, 17, 19, 23, 34, 36, 12, 1, 9, 0, 8, 5, 4, 22, 48, 49

**Table 16 ijerph-19-12272-t016:** Comparison between the best hybridizations of classification and feature selection techniques with the training and testing datasets of each data subset.

Dataset	Quality Metrics					Hybridization Classification Technique + Feature Selection
	FP Rate	Precision	Recall	F-Measure	ROC Area	
**Aruba CASAS** **–** **raw**	0.60%	95.20%	94.90%	94.90%	99.70%	LMT + Gain Ratio (27 Features)
	**0** **.** **50%**	**95** **.** **10%**	**94** **.** **90%**	**94** **.** **90%**	**99** **.** **70%**	**LMT + Gain Ratio (24** **Features** **)**
**Aruba—duration**	0.50%	95.80%	95.80%	95.80%	99.20%	JRIP + One R (47 Features)
	**0** **.** **60%**	**95** **.** **90%**	**95** **.** **90%**	**95** **.** **80%**	**99** **.** **70%**	**LMT + One R (33** **Features** **)**
**Aruba—sensor based**	0.50%	95.10%	94.90%	94.80%	99.70%	LMT + Info Gain (47 Features)
	**0** **.** **60%**	**95** **.** **10%**	**94** **.** **90%**	**94** **.** **80%**	**99** **.** **70%**	**LMT + Gain Ratio (31** **Features** **)**

**Table 17 ijerph-19-12272-t017:** Attributes with the highest incidence in the classification of the LMT technique identified with the One R feature selection technique for the Aruba CASAS–duration dataset.

ID	Attribute	Priority	ID	Attribute	Priority	ID	Attribute	Priority
1	M018	70.12	12	D004-close	51.97	23	M026	47.49
2	M019	69.94	13	D004-open	51.80	24	M004	47.43
3	M015	69.08	14	M030	51.49	25	T001	47.36
4	M009	68.30	15	M007	51.18	26	M022	47.34
5	M017	68.12	16	M003	50.89	27	M005	47.25
6	duration	66.02	17	M020	50.78	28	M027	47.03
7	M016	65.04	18	M002	50.58	29	T005	46.96
8	M013	64.17	19	T003	48.58	30	M028	46.85
9	M014	59.61	20	M029	47.98	31	M008	46.52
10	M021	55.99	21	T004	47.65	32	M006	45.94
11	events	52.15	22	T002	47.63	33	M023	45.87

**Table 18 ijerph-19-12272-t018:** LMT classification results + Gain Ratio with cross-validation with 10 folds for Aruba CASAS–raw dataset.

Class	LMT + Gain Ratio (24 Features)				
	FP Rate	Precision	Recall	F-Measure	ROC Area
**Sleeping**	0.00%	99.20%	99.60%	99.40%	100.00%
**Bed_to_Toilet**	0.00%	98.20%	100.00%	99.10%	100.00%
**Meal_Preparation**	0.30%	99.20%	99.00%	99.10%	99.50%
**Relax**	0.80%	99.10%	99.40%	99.20%	99.70%
**Housekeeping**	0.10%	86.40%	82.60%	84.40%	96.40%
**Eating**	0.10%	98.30%	96.10%	97.20%	99.80%
**Leave_Home**	2.00%	65.20%	53.40%	58.70%	97.20%
**Enter_Home**	3.30%	63.20%	73.40%	67.90%	97.50%
**Work**	0.10%	97.50%	97.50%	97.50%	100.00%
**Average**	0.80%	94.10%	**94** **.** **10%**	94.10%	**99** **.30%**

**Table 19 ijerph-19-12272-t019:** LMT + One R classification results in cross-validation with 10-folds of the Aruba CASAS–duration dataset.

Class	LMT + One R (33 Features)				
	FP Rate	Precision	Recall	F-Measure	ROC Area
**Sleeping**	0.00%	99.60%	99.60%	99.60%	99.80%
**Bed_to_Toilet**	0.00%	99.10%	100.00%	99.60%	100.00%
**Meal_Preparation**	0.40%	98.80%	98.60%	98.70%	99.60%
**Relax**	0.80%	99.10%	99.30%	99.20%	99.70%
**Housekeeping**	0.20%	66.70%	69.60%	68.10%	94.90%
**Eating**	0.10%	97.10%	93.40%	95.20%	97.70%
**Leave_Home**	2.80%	62.80%	66.80%	64.70%	97.10%
**Enter_Home**	2.30%	68.00%	63.30%	65.60%	97.20%
**Work**	0.20%	94.50%	99.20%	96.80%	100.00%
**Average**	0.80%	94.10%	**94** **.** **10%**	94.00%	**99** **.20%**

**Table 20 ijerph-19-12272-t020:** LMT classification results + Gain Ratio with cross-validation with 10-folds of the Aruba CASAS–sensor-based dataset.

Class	LMT + Gain Ratio (31 Features)				
	FP Rate	Precision	Recall	F-Measure	ROC Area
**Sleeping**	0.10%	98.90%	100.00%	99.40%	100.00%
**Bed_to_Toilet**	0.10%	97.40%	100.00%	98.70%	100.00%
**Meal_Preparation**	0.40%	98.80%	99.00%	98.90%	99.70%
**Relax**	0.70%	99.20%	99.20%	99.20%	99.40%
**Housekeeping**	0.10%	86.40%	82.60%	84.40%	90.70%
**Eating**	0.00%	99.40%	94.50%	96.90%	99.80%
**Leave_Home**	2.10%	64.90%	53.40%	58.60%	97.30%
Enter_Home	3.30%	63.10%	73.40%	67.80%	97.70%
Work	0.10%	95.90%	97.50%	96.70%	99.90%
Average	0.80%	94.00%	**94** **.** **00%**	93.90%	**99** **.30%**

**Table 21 ijerph-19-12272-t021:** LMT classification results + Gain Ratio with cross-validation with 10 folds for Aruba CASAS–sensor-based dataset.

Dataset	Quality Metrics					Hybridization Classification Technique + Feature Selection (10-Fold Cross Validation)
	FP Rate	Precision	Recall	F-Measure	ROC Area	
Aruba CASAS–*raw*	0.80%	94.10%	**94** **.** **10%**	94.10%	99.30%	LMT + Gain Ratio (24 Features)
Aruba CASAS–*duration*	0.80%	94.10%	**94** **.** **10%**	94.00%	99.20%	LMT + One R (33 Features)
Aruba CASAS–*sensor based*	0.80%	94.00%	**94** **.** **00%**	93.90%	99.30%	LMT + Gain Ratio (31 Features)

**Table 22 ijerph-19-12272-t022:** Statistical analysis—ANOVA.

Models	F	Probability	The Critical Value for F
M1 (LMT + Gain Ratio 24 Features) vs. M2 (LMT + One R 33 Features)	0.058300716	0.812269355	4.493998478
M1 (LMT + Gain Ratio 24 Features) vs. M3 (LMT + Info Gain 47 Features)	0.034326866	0.855341542	4.493998478
M2 (LMT + One R 33 Features) vs. M3 (LMT + Info Gain 47 Features)	0.00182054	0.966494302	4.493998478

**Table 23 ijerph-19-12272-t023:** Comparison between the best hybridization of classification techniques and feature selection with training and testing for each dataset.

Class	Aruba CASAS–Raw (LMT + Gain Ratio 24 Features)		Aruba CASAS–Duration (LMT + One R 33 Features)		Aruba CASAS–Sensor-based (LMT + Gain Ratio 31 Features)	
	Recall	ROC Area	Recall	ROC Area	Recall	ROC Area
Sleeping	100.00%	100.00%	100.00%	100.00%	100.00%	100.00%
Bed_to_Toilet	100.00%	100.00%	97.80%	100.00%	100.00%	100.00%
Meal_Preparation	98.50%	99.80%	98.60%	99.90%	98.30%	100.00%
Relax	99.40%	99.90%	99.70%	99.80%	99.80%	100.00%
Housekeeping	100.00%	100.00%	77.80%	100.00%	88.90%	100.00%
Eating	100.00%	100.00%	98.60%	98.80%	98.60%	100.00%
Leave_Home	55.60%	98.10%	83.50%	98.50%	55.60%	98.10%
Enter_Home	75.90%	97.90%	62.50%	98.20%	75.90%	97.90%
Work	100.00%	100.00%	96.00%	100.00%	100.00%	100.00%
Average	94.90%	99.70%	**95** **.** **90%**	**99** **.** **70%**	94.90%	99.70%

## Data Availability

Public Dataset Casas Aruba, Available: http://casas.wsu.edu/datasets/.

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
