# Peer review of "Predictive Model for Human Activity Recognition Based on Machine Learning and Feature Selection Techniques"

_ijerph, 2022, doi:10.3390/ijerph191912272_

Round 1
Reviewer 1 Report
The topic that is discussed in this paper is very important. Quite a lot of fundamental details were also provided by the authors. However, this paper lacks of focus and innovation. It didn't address the problems to be solved or propose any innovative ideas. It is just a report of results of experimental exercises. This is not suitable for publications at this high-profile journal.
Author Response
Thank you for your comments

Reviewer 2 Report
-
Line 39-41: Please redraft the sentence. It's difficult to understand what authors want to convey.
-
For section 2.2, please provide a supplementary table describing the list of techniques available to generate HAR datasets and potential scenarios where the dataset is applicable.
-
Line 153-158. Please remove the section. This is unnecessary text, adding no value.
-
Line 159-170. Please include this information in supp. Table (comment-2 above), and remove this section as it is similar to section 2.2.
-
Line 171-176. Please elaborate, why these datasets were generated and how these datasets can be accessed.
-
It's not clear, what the total number of features is and the data points generated in CASAS datasets are.
-
Line 210-227. Please remove this section. It's just adding bulk information with no value.
-
Please clearly explain what feature selection method was used and what the outcome is i.e. most relevant features.
-
What is the size of the train and test data set? Did the authors also include an independent validation dataset, for each of the experimental scenarios?
-
The manuscript is poorly organized and written. It's difficult to understand how different sections are inter-connected. I recommend authors to rewrite the manuscript and keep only the relevant information for the readers and reorganize different sections of the manuscript to keep relevant information only, i.e. dataset description, feature selection, and classification models used. I liked section 4.1 and I believe it is the core of the study, and it needs to be broken down into different sub-section with sub-headings.
Author Response
Dear reviewer,
Thanks for the recomendations.

Reviewer 3 Report
This is an interesting paper about using Machine Learning models and Feature Selection techniques on Recognition of Human Activities.
With well designed experiments, the authors measures different machine learning models performance in terms of prediction precision and recall, and conclude the best model on the given dataset. The authors also apply some feature selection techniques to achieve similar or better prediction performance with less features.
It is well-organized and clearly describes the design of the experiment setup. The authors also provide solid experiment proofs on different models' performance.
Overall this is a good paper for IJERPH and I would suggest for accepting.
Author Response
Thank you for your comments

Round 2
Reviewer 2 Report
Dear Editors,
Thank you for considering me to review the manuscript again. I reviewed the revised manuscript and the manuscript looks in better form. However, my concern is the presentation of results and, to me, it is still difficult to conclude and receive a take-home message/product from this article.
Given the wide range of online readers, I recommend this paper for publication, only after the authors recheck the manuscript for all the potential grammatical mistakes.
I thank you again for considering me to review this manuscript.
Best,
Abhinav Kaushik
Author Response
Thanks you for the reviewers
